# Probabilistic Group Mask Guided
# Discrete Optimization for Incremental Learning

**Fengqiang Wan**[1]   **Yang Yang**[1]

## Abstract

Incremental learning (IL) aims to sequentially learn new tasks while mitigating catastrophic forgetting. Among various IL strategies, parameter-isolation methods stand out by using mask techniques to allocate distinct parameters to each task, explicitly addressing forgetting. However, existing approaches often disregard parameter dependencies, resulting in an over-reliance on newly allocated parameters. To address this issue, we propose Probabilistic Group Mask selection (PGM), a group-wise approach that captures parameter dependencies by exploring candidate masks within each group. Specifically, PGM partitions parameters into groups with multiple candidate masks, assigning probabilities to these masks and leveraging Gumbel-Softmax for differentiable sampling, enabling efficient optimization of the discrete mask selection process. Our theoretical analysis demonstrates that incorporating parameter dependencies enhances sub-network selection. Experiments conducted on standard benchmarks confirm its superior effectiveness compared to existing IL approaches. The source code is available at: `https://github.com/njustkmg/ICML25-PGM`.

## 1. Introduction

Incremental Learning (IL) enables models to incrementally acquire new knowledge, making it highly applicable to domains such as autonomous driving (Fang et al., 2024a) and healthcare (Lesort et al., 2020). A good IL model is expected to keep the memory of all seen tasks upon learning new knowledge (Hocquet, 2021; Yang et al., 2023a). However, due to the limited access to previous data, the learning phase is naturally sensitive to the current task, hence result-

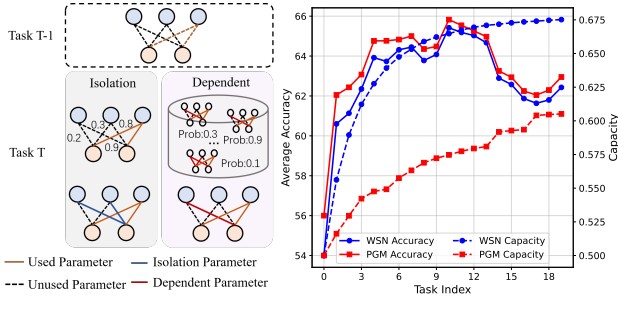

(a) Isolation vs. Dependency    (b) Performance

*Figure 1.* Illustration of different parameter selection strategies. (a) Isolation-based methods evaluate parameters independently and select the top-ranked ones, while dependency-aware methods consider parameter interactions, enabling joint selection based on their collective contributions. (b) The dependency-aware approach, exemplified by PGM, demonstrates enhanced ACC and CAP compared to isolation-based methods like WSN, underscoring the advantages of incorporating parameter dependencies.

ing in a major challenge in IL called Catastrophic Forgetting (CF) (Lange et al., 2022; Zhou et al., 2022), which refers to the drastic performance drop on past knowledge after learning new knowledge.

Several promising approaches have been proposed for IL that focus on mitigating CF (Hung et al., 2019; Kumar et al., 2021; Yoon et al., 2018a; Liu et al., 2024; Yang et al., 2019; 2023b). Among these, parameter isolation achieves complete prevention of catastrophic forgetting (Wortsman et al., 2020; Serrà et al., 2018; Hu et al., 2024), by masking distinct sub-networks for each task within a shared model architecture. However, a key challenge lies in determining the optimal allocation of parameters for each task, balancing the preservation of prior knowledge with adaptation to new tasks. Several approaches rely on handcrafted criteria to assess parameter importance, such as the absolute magnitude of the weights (Mallya & Lazebnik, 2018) or gradient information (Konishi et al., 2023), but these proxies lead to suboptimal allocations due to not considering their true contributions to the task. Alternatively, parameters can be assigned learnable weights (Mallya et al., 2018; Kang et al., 2022; Hu et al., 2024), which are updated via

---

[1]Nanjing University of Science and Technology. Correspondence to: Yang Yang <yyang@njust.edu.cn>.

*Proceedings of the 42nd International Conference on Machine Learning*, Vancouver, Canada. PMLR 267, 2025. Copyright 2025 by the author(s).

gradients during training to more effectively capture their true importance, enabling more precise parameter allocation. However, existing parameter selection methods, as illustrated in the left part of Figure 1a, independently score and rank each parameter, offering simplicity but neglecting critical dependencies among them. As established in Theorem 3.2, neglecting parameter dependencies and their collective contributions leads to suboptimal sub-network. In contrast, dependency-aware methods, shown in the right part of Figure 1a, explicitly account for parameter dependencies, facilitating joint selection that optimizes their combined contributions. According to Definition 3.1, incorporating these dependencies throughout the learning sequence enhances parameter reuse, reducing the need for newly introduced parameters in each task. As demonstrated in Figure 1b, this dependency-aware reuse enables each task to achieve comparable or superior performance with fewer additional parameters, resulting in more compact sub-networks. This reduction in model expansion preserves more capacity for future tasks, facilitating better adaptability to unseen tasks while maintaining scalability.

Therefore, we propose Probabilistic Group Mask selection (PGM), which models the mask selection process as a probabilistic sampling process. Specifically, PGM organizes parameters into groups, with each group containing multiple candidate masks, and assigns a probability to each candidate mask. This probabilistic sampling process is reparameterized using the Gumbel-Softmax technique (Jang et al., 2017), enabling a deterministic approximation of the selection process and facilitating end-to-end optimization through gradient. Our theoretical analysis demonstrates that incorporating parameter dependencies improves subnetwork selection by capturing synergies between parameters, leading to more effective utilization of model capacity. Extensive experiments on widely-used datasets demonstrate that our method achieves competitive performance.

## 2. Preliminaries

Task-incremental learning (TIL) is a key paradigm in incremental learning, where the task identity $t$ is available during the inference phase. This enables the model to specialize its processing to each task as it learns sequentially from a stream of tasks $\{T_1, T_2, \cdots, T_N\}$. For each task $T_t$, the model $\mathcal{F}(\cdot, \theta)$, with parameters $\theta$, learns from the dataset $D_t = \{X_t, Y_t\}$, where $X_t$ is the input feature set and $Y_t$ is the corresponding label set. The objective for task $t$ is to minimize the task-specific loss, expressed as:

$$\theta^* = \arg\min_{\theta} \frac{1}{n_t} \sum_{i=1}^{n_t} \mathcal{L}(\mathcal{F}(x_i^t; \theta), y_i^t),$$

where $\mathcal{L}$ represents the loss function and $n_t$ is the number of samples in $D_t$.

In TIL, parameter isolation methods have proven to be highly effective in mitigating catastrophic forgetting. These methods leverage the over-parameterization of neural networks to create task-specific subnetworks, isolating the parameters dedicated to each task. This approach minimizes interference between tasks and preserves performance on previously learned tasks, while still allowing for flexibility in learning future tasks. Mathematically, for a given set of model parameters $\theta$, a binary mask $m_t^*$ defines the subnetwork for task $t$, constrained by $\|m_t^*\| \leq c$, where $c$ represents the subnetwork's capacity. The optimization problem is formulated as:

$$m_t^* = \arg\min_{m_t \in \{0,1\}^{|\theta|}} \frac{1}{n_t} \sum_{i=1}^{n_t} \mathcal{L}(\mathcal{F}(x_i^t; \theta \odot m_t), y_i^t)$$
$$- \mathcal{L}(\mathcal{F}(x_i^t; \theta), y_i^t) \quad \text{s.t. } \|m_t^*\| \leq c,$$
$$\tag{1}$$

where $\mathcal{L}$ denotes the task-specific loss function, $\odot$ denotes elementwise multiplication. The goal is to find the optimal mask $m_t^*$ that isolates the task-specific subnetwork, ensuring that it achieves task performance comparable to the full network, while adhering to the capacity constraint.

## 3. Method

### 3.1. Theoretical Understanding

To enhance subnetwork selection, we propose a group-wise strategy that incorporates parameter dependencies, improving both parameter reuse and error reduction. By structuring the parameter selection process into groups, this approach facilitates the joint evaluation of parameter dependencies within each group, leading to more accurate and effective subnetwork optimization.

**Definition 3.1** (Parameter Reuse with Dependency). Let $K$ be the set of parameters, and let parameter dependencies that influence their reuse across tasks. The total parameter reuse $R$ across $N$ tasks is defined as:

$$R = \sum_{i=1}^{|\theta|} \mathbb{I}\left(\sum_{t=1}^{N} m_{t,i} > 0\right),$$

where $\mathbb{I}(\cdot)$ is the indicator function, and $m_{t,i}$ denotes the effective mask for parameter $\theta_i$ in task $t$, which is determined by the joint contribution of $K$ parameters:

$$m_{t,i} = \mathbb{I}\left(\left(s_{t,i} + \sum_{j \in \mathcal{N}_i^K} \rho_{j,i}\right) > \tau\right).$$

In this definition, $s_{t,i}$ represents the independent score of parameter $\theta_i$ for task $t$, $\rho_{j,i}$ quantifies the dependency between parameter $\theta_i$ and $\theta_j$, and $\mathcal{N}_i^K$ is the set of $K - 1$ parameters most strongly interacting with $\theta_i$. The threshold $\tau$ determines the selection criterion for parameter reuse.

According to Definition 3.1, when $K = 1$, the dependency term is absent, resulting in parameter selection based solely on $s_{t,i}$, which corresponds to independent selection. In contrast, when $K > 1$, the joint evaluation of dependencies with $K - 1$ additional parameters increases the probability of reusing $\theta_i$ across tasks, thereby achieving a higher $R$.

**Theorem 3.2** (Error Reduction via Dependency). *Dividing $N$ parameters into $N/K$ groups of size $K$ reduces the uncertainty in parameter selection by leveraging local interactions within each group. The probability of an incorrect selection for a parameter combination $\mathcal{C}$ under the groupwise strategy is given by:*

$$P_{wrong}(\mathcal{C}) = \prod_{b=1}^{N/K} \Phi\left( -\frac{\sum_{i \in \mathcal{S}_b^*} TS(\theta_i) - \sum_{i \in \mathcal{C}_b} TS(\theta_i)}{\sqrt{K}\sigma} \right),$$

*where $\mathcal{S}_b^*$ denotes the ideal parameter combination within group $b$, $\mathcal{C}_b$ represents the selected parameter combination, $TS(\cdot)$ is the ideal score, $\sigma$ is the standard deviation of the noise, and $\Phi(\cdot)$ denotes the cumulative distribution function.*

By explicitly modeling parameter dependencies, the groupwise approach reduces evaluation error by capturing interactions that contribute to task-specific performance. Theorem 3.2 demonstrates that incorporating dependencies into the selection process lowers the likelihood of errors by ensuring a more informed evaluation. Structuring parameters into groups facilitates this joint consideration, avoiding the limitations of independent evaluations and leading to more accurate subnetwork selection. The detailed proof can be found in the Appendix A.

### 3.2. Probabilistic Group Mask Selection

Building on the theoretical results above, the challenge of dividing $K$ parameters into groups is reframed as the task of selecting optimal subsets within each group to achieve specific objectives. This method introduces parameter grouping, reformulates the problem as a sampling process, and utilizes Gumbel-Softmax for efficient optimization. Furthermore, an adaptive initialization leverages the transferability of prior task distributions to facilitate task-specific adaptations.

**Parameter Group:** Consider a parameter group consisting of $K$ parameters, denoted as $\mathcal{W} \in \mathbb{R}^{1 \times K}$. The goal is to determine the optimal binary mask $\mathcal{M}^* \in \mathcal{B}^{1 \times K}$ of the same dimension. This leads to a discrete candidate set $\mathcal{S}^K$, which contains $\sum_{j=1}^{K-1} \binom{K}{j}$ possible masks, expressed as:

$$\mathcal{S}^K = \{\mathcal{M}^* \in \mathcal{B}^{1 \times K}\} = \bigcup_{j=1}^{K-1} \{\mathcal{M}_{j,k}\},$$

where $\{\mathcal{M}_{j,k}\}$ denotes the set of masks with exactly $j$ nonzero elements, and $k$ indexes the specific subsets. The

model comprises numerous parameter groups, denoted as $\theta_l$, each necessitating the selection of a corresponding mask $M_l$. To identify an optimal subnetwork, Equation (1) can be reformulated as:

$$\{M_l^*\}_t = \underset{\{M_l | M_l \in \mathcal{S}^K\}}{\arg\min} \frac{1}{n_t} \sum_{i=1}^{n_t} \Big[ \mathcal{L}\big(\mathcal{F}(x_i^t; \theta \odot \{M_l\}), y_i^t\big)$$
$$- \mathcal{L}\big(\mathcal{F}(x_i^t; \theta), y_i^t\big) \Big], \tag{2}$$

where the operator $\odot$ represents elementwise multiplication.

Identifying the optimal set of masks $\{M_l^*\}$ poses significant challenges due to the nondifferentiable nature of mask selection and the high dimensionality of model parameters. To address these difficulties, the mask selection problem is reformulated as a sampling-based process, thereby enabling efficient optimization while preserving the capacity for task-specific adaptability.

**Sampling Process:** Let $\mathcal{W} \in \mathbb{R}^{1 \times K}$ denote a parameter groups containing $K$ parameters. Identifying the exact optimal mask for such a group is inherently complex due to interdependencies with other parameter groups. To address this, an independent sampling approach is employed for each group, providing a tractable and efficient means of evaluating overall model quality (Fang et al., 2024b). Empirical studies further demonstrate that model performance tends to stabilize when $K$ surpasses a certain threshold, highlighting the scalability of the proposed sampling strategy.

To enable efficient sampling of $M$, two categorical distributions are defined: $p_1, p_2, \ldots, p_j$, which determine the number of nonzero elements, and $q_{j1}, q_{j2}, \ldots, q_{jk}$, which select a specific mask from the candidate set. These distributions are normalized such that $\sum_j p_j = 1$ and $\sum_k q_{jk} = 1$. Masks that demonstrate strong performance are assigned higher probabilities, guiding subsequent sampling toward promising candidates. Through iterative sampling and probability updates, the process converges to a distribution that prioritizes high-performing masks, thereby increasing the likelihood of discovering an optimal subnetwork. This reformulates Equation (2) as a probabilistic sampling problem:

$$\{q^*(M_{jk} \mid j)\}_t = \underset{p(j), q(M_{jk}|j)}{\arg\min} \frac{1}{n_t} \sum_{i=1}^{n_t} \mathbb{E}_{j \sim p(j), M_i \sim q(M_{jk}|j)}$$
$$\Big[ \mathcal{L}\big(\mathcal{F}(x_{i,t}; \theta \odot \{M_{jk}\}), y_{i,t}\big) - \mathcal{L}\big(\mathcal{F}(x_{i,t}; \theta), y_{i,t}\big) \Big]. \tag{3}$$

Here, $p(j)$ defines the probability distribution over the number of nonzero elements in a mask, and $q(M_{jk} \mid j)$ represents the conditional probability of selecting a specific mask $M_{jk}$ given $j$. While gradient descent can be applied to optimize objectives when gradients are available, the inherently nondifferentiable nature of categorical sampling

presents significant challenges. This issue arises because sampling from a categorical distribution involves discrete operations, which lack the gradient information required for optimization, thereby impeding the direct use of standard optimization techniques.

**Differentiable Sampling of Masks:** To address the two-stage sampling problem involving $p(j)$ and $q(M_{jk} \mid j)$, the Gumbel-Softmax reparameterization technique is employed. This method facilitates differentiable sampling by providing a continuous approximation to the discrete process. Specifically, for $p(j)$, the discrete sampling is relaxed into the following differentiable form:

$$\tilde{p}(j) = \frac{\exp((\log(p(j)) + g_j)/\tau_1)}{\sum_{j'} \exp((\log(p(j')) + g_{j'})/\tau_1)},$$

where $g_j$ are independent samples from the Gumbel distribution $-\log(-\log(U))$, with $U \sim \text{Uniform}(0, 1)$, and $\tau_1 > 0$ is a temperature parameter that controls the degree of approximation. As $\tau_1 \to 0$, $\tilde{p}(j)$ approaches a one-hot vector. Similarly, the sampling of a specific mask $M_{jk}$ for $q(M_{jk} \mid j)$ can be relaxed using the same reparameterization technique. The relaxed form is expressed as:

$$\tilde{q}(M_{jk} \mid j) = \frac{\exp((\log(q(M_{jk} \mid j)) + g_k)/\tau_2)}{\sum_{k'} \exp((\log(q(M_{j,k'} \mid j)) + g_{k'})/\tau_2)},$$

where $g_k$ introduces stochasticity, and $\tau_2$ controls the smoothness of mask selection. These reparameterized forms ensure that the sampling process remains differentiable, enabling efficient optimization of $p(j)$ and $q(M_{jk} \mid j)$ using gradient-based methods.

Using these reparameterized distributions, the final differentiable mask $\tilde{M}$ is constructed by integrating the soft selection over $j$ with the conditional distribution over masks. This results in a weighted combination of the candidate masks, expressed as:

$$\tilde{M} = \sum_j \tilde{p}(j) \sum_k \tilde{q}(M_{jk} \mid j) M_{jk}. \tag{4}$$

The reparameterization ensures that the sampling process is fully differentiable, allowing $p(j)$ and $q(M_{jk} \mid j)$ to be optimized through standard gradient-based methods. Instead of directly optimizing these probabilities, their corresponding logits $\pi_j$ and $\phi_j^k$ are learned. Scaling factors $\kappa_1$ and $\kappa_2$ are applied to adjust the sharpness of the distributions, with the probabilities computed as follows:

$$p(j) = \frac{\exp(\pi_j \cdot \kappa_1)}{\sum_{j'} \exp(\pi_{j'} \cdot \kappa_1)}, q(M_{jk}|j) = \frac{\exp(\phi_j^k \cdot \kappa_2)}{\sum_{k'} \exp(\phi_j^{k'} \cdot \kappa_2)}.$$

This formulation enables the construction of the differentiable mask $\tilde{M}$, efficiently addressing the sampling problem outlined in Equation (3). By balancing computational

efficiency and flexibility, this approach facilitates precise mask selection while maintaining scalability. For task $t$, the optimal mask is obtained by selecting the combination of $j$ and $k$ that maximizes the joint probability of $p(j)$ and $q(M_{jk} \mid j)$, formally expressed as:

$$M_t^* = \{M_{jk} \mid \arg\max p(j) q(M_{jk} \mid j)\}. \tag{5}$$

By design, the mask selection process adapts dynamically to task-specific requirements, focusing on the most relevant parameter subsets. This adaptability not only enhances interpretability but also reduces the risk of suboptimal selections, ensuring effective utilization of model capacity across tasks.

**Task-Informed Mask Initialization:** To effectively initialize the probability distributions $p(j)$ and $q(M_{jk} \mid j)$ for task $t$, this method incorporates the similarities between the current task mask and those of all previous tasks $(M_1, M_2, \ldots, M_{t-1})$. By leveraging these similarities, the initialization process allows previously learned knowledge to directly influence the sampling process. This approach not only facilitates efficient parameter reuse but also ensures that the model retains adaptability to task-specific requirements. The contribution of each prior task is weighted based on its similarity to the current task, thereby shaping the probability distributions in a task-aware manner. To quantify this influence, the similarity between current task mask $\hat{M}_t$ and prior task mask $M_n$ ($n \in \{1, \ldots, t-1\}$) is computed as:

$$\text{sim}(M_n, \hat{M}_t) = M_n \hat{M}_t^\top - \frac{1}{|S|} \sum_i (M_i \hat{M}_t^\top).$$

In this formulation, the similarity is measured using the inner product between $M_n$ and $\hat{M}_t$, with an adjustment to remove the mean similarity across all candidate masks. This adjustment is critical as it normalizes the similarities, reducing the influence of global variations across the mask space and focusing on meaningful relationships between the current and prior tasks. Building on this similarity measure, the initialization process prioritizes candidate masks with higher similarity to previous tasks by adjusting their probabilities accordingly. Specifically, the logits for $p(j)$ and $q(M_{jk} \mid j)$ are updated using the aggregated similarity:

$$\pi_t' = \pi_t + \sum_{n=1}^{t-1} \text{sim}(M_n, \hat{M}_t) \cdot \sigma(\pi_n) \cdot \alpha_1,$$

$$\phi_t' = \phi_t + \sum_{n=1}^{t-1} \text{sim}(M_n, \hat{M}_t) \cdot \sigma(\phi_n) \cdot \alpha_2.$$

Here, $\sigma(\pi_n)$ and $\sigma(\phi_n)$ denote the standard deviations of the logits for task $n$, and the hyperparameters $\alpha_1$ and $\alpha_2$ determine the extent to which prior task information influences the current initialization. By incorporating both the similarity and variability of prior task distributions, the initialization process achieves a balance between knowledge transfer and task-specific differentiation.

*Table 1.* Performance comparison of PGM with other IL methods across three datasets. Under the same conditions, ACC is prioritized over BWT, as it reflects the overall balance between network stability and plasticity, whereas BWT solely quantifies the degree of forgetting.

| Method | CIFAR-100 Split | | | CIFAR-100 Superclass | | | TinyImageNet | | |
|---|---|---|---|---|---|---|---|---|---|
| | ACC (%) ↑ | CAP (%)↓ | BWT (%) ↑ | ACC (%) ↑ | CAP (%)↓ | BWT (%) ↑ | ACC (%) ↑ | CAP (%)↓ | BWT (%) ↑ |
| Multitask | 79.75 (±0.38) | 100.0 | - | 61.00 (±0.20) | 100.0 | - | 77.10 (±1.06) | 100.0 | - |
| La-MaML | 71.37 (±0.67) | 100.0 | -5.39 (±0.53) | 54.44 (±1.36) | 100.0 | -6.65 (±0.85) | 66.90 (±1.65) | 100.0 | -9.13 (±0.99) |
| GPM | 73.18 (±0.52) | 100.0 | -1.17 (±0.27) | 57.33 (±0.37) | 100.0 | -0.37 (±0.12) | 67.39 (±0.47) | 100.0 | -1.45 (±0.22) |
| FS-DGPM | 74.33 (±0.41) | 100.0 | -2.71 (±0.21) | 58.81 (±0.34) | 100.0 | -2.97 (±0.35) | 70.41 (±0.87) | 100.0 | -2.11 (±0.84) |
| RP2F | 75.89 (±0.45) | 100.0 | -0.82 (±0.13) | 60.11 (±0.23) | 100.0 | -0.32 (±0.02) | 70.32(±0.29) | 100.0 | -1.05 (±0.43) |
| PackNet | 72.39 (±0.37) | 96.38 (±0.38) | 0.0 | 58.78 (±0.52) | 126.65 (±0.00) | 0.0 | 55.46 (±1.22) | 188.67 (±0.00) | 0.0 |
| SupSup | 75.47 (±0.33) | 129.00 (±0.00) | 0.0 | 61.70 (±1.31) | 162.49 (±0.00) | 0.0 | 59.60 (±1.05) | 214.52 (±0.89) | 0.0 |
| WSN | 76.38 (±0.34) | 99.13 (±0.48) | 0.0 | 61.79 (±0.23) | 80.93 (±1.58) | 0.0 | 69.06 (±0.82) | 92.03 (±1.80) | 0.0 |
| SPG | 77.12(±0.42) | 90.24 (±0.89) | -1.02 (±0.38) | 62.04 (±0.39) | 75.93 (±0.97) | -0.53 (±0.17) | 70.16 (±0.74) | 85.53 (±1.19) | -1.26 (±0.17) |
| PGM | **77.29(±0.31)** | **63.51(±0.98)** | **0.0** | **62.56(±0.71)** | **50.38(±1.05)** | **0.0** | **70.99(±0.54)** | **80.26(±0.86)** | **0.0** |

## 4. Experiments

We validate our method across multiple benchmark datasets, comparing its performance against relevant incremental learning baselines. For all experiments presented in this paper, we utilize TIL with a multi-head configuration. The experimental setups are carefully aligned with those employed in recent works (Kang et al., 2022). The evaluation encompasses comprehensive performance comparisons, ablation studies examining module effectiveness, group size variations, and adaptability to different training paradigms, as well as an in-depth analysis of computational efficiency, parameter dependencies, and parameter distribution.

### 4.1. Setup

**Datasets and Evaluation Metrics:** We use three different popular datasets, including Split CIFAR-100 (Krizhevsky & Hinton, 2009), CIFAR-100 Superclass (Yoon et al., 2018b), Split TinyImageNet (Krizhevsky et al., 2017). To evaluate IL methods, we evaluate all methods on three metrics: ACC, CAP, BWT. ACC measures the average classification performance across all tasks, defined as ACC $= \frac{1}{T}\sum_{i=1}^{T} A_{T,i}$, where $A_{T,i}$ represents the test accuracy for task $i$ after training on task $T$. CAP quantifies the proportion of nonzero weights and prime masks used for all tasks, calculated as CAP $= (1 - S) + (1 - \alpha)T \times \frac{1}{32}$, where $S$ is the sparsity of $M_T$ and $\alpha$ (approximately 0.78) is the average compression rate achieved through 7-bit Huffman encoding. Finally, BWT assesses the degree of forgetting in incremental learning scenarios, expressed as BWT $= \frac{1}{T-1}\sum_{i=1}^{T-1}(A_{T,i} - A_{i,i})$.

**Baselines:** To thoroughly evaluate the performance of the proposed PGM, a comparison is conducted with several prevalent incremental learning baselines. Specifically, (1) parameter isolation methods, including PackNet (Mallya & Lazebnik, 2018), SupSup (Wortsman et al., 2020),and WSN (Kang et al., 2022), SPG (Konishi et al., 2023); (2) parameter

regularization techniques such as La-MAML (Joseph & Gu, 2021), GPM (Saha et al., 2021), FS-DGPM (Deng et al., 2021), RP2F (Sun et al., 2024), and MTD (Wen et al., 2024); and (3) the naive sequential training strategy, referred to as FINETUNE, in addition to multitask learning (MTL), which serve to establish lower and upper bounds for performance.

**Implementation Details:** To ensure a fair comparison, we follow the experimental protocols from (Kang et al., 2022), maintaining consistent backbone architectures across datasets. For instance, we use a modified AlexNet for Split CIFAR-100 and a customized LeNet for CIFAR-100 Superclass. Training employs the Adam optimizer with a momentum of 0.9, with each task trained for a fixed number of epochs to ensure convergence. Experiments are conducted using PyTorch on a high-performance computing platform with NVIDIA 4090 GPUs. Additional hyperparameter settings are provided in the Appendix B.

### 4.2. Main Results

**Overall Performance:** The performance of the proposed PGM is rigorously compared with several state-of-the-art methods across three key evaluation metrics on three prominent benchmark datasets, as presented in Table 1. PGM consistently outperforms all existing approaches, achieving the highest ACC of 77.29%, 62.56%, and 70.99%, respectively. These results demonstrate the efficacy of PGM in addressing the challenges associated with IL. In particular, as highlighted in Table 1, (1) In comparison to parameter regularization methods, both PGM and parameter isolation approaches exhibit zero forgetting, effectively preventing catastrophic forgetting across tasks. Notably, PGM achieves a superior ACC, suggesting that it not only mitigates forgetting but also enhances overall model performance; (2) When compared to parameter isolation methods, PGM achieves better ACC with a smaller set of parameters, indicating a more efficient use of model resources. As shown in Figure 2a, PGM outperforms WSN in terms of accuracy across most tasks,

*Table 2.* Results of ablation studies highlighting the contributions of the Group and Mask Initialization modules across CIFAR-100 Split, CIFAR-100 Superclass, and TinyImageNet datasets. The table reports accuracy (ACC) and capacity (CAP) metrics, demonstrating the impact of each component on performance and resource utilization.

| Group | Mask Initialization | CIFAR-100 Split | | CIFAR-100 Superclass | | TinyImageNet | |
|---|---|---|---|---|---|---|---|
| | | ACC (%) ↑ | CAP (%) ↓ | ACC (%) ↑ | CAP (%) ↓ | ACC (%) ↑ | CAP (%) ↓ |
| ✗ | ✗ | 76.38 | 99.13 | 61.79 | 80.93 | 69.06 | 92.03 |
| ✓ | ✗ | 76.98 | 65.97 | 62.04 | 54.25 | 70.13 | 82.39 |
| ✓ | ✓ | **77.29** | **63.51** | **62.56** | **50.38** | **70.99** | **80.26** |

further reinforcing the effectiveness of the approach; (3) Furthermore, Figure 2b illustrates that, throughout the entire training process, PGM exhibits a higher degree of overlap in model parameters between tasks, leading to more efficient knowledge transfer. This results in a reduction in the dependence on newly introduced parameters, in contrast to WSN, which shows a greater reliance on task-specific parameters. Such behavior suggests that PGM is more adept at retaining and reusing previously learned knowledge, thereby improving its ability to generalize across tasks while minimizing the cost of learning new tasks.

### 4.3. Ablation study

**Effectiveness of Each Component:** Table 2 provides a detailed summary of the ablation studies conducted across multiple datasets, highlighting the individual and combined contributions of the Group and Mask Initialization modules to the overall performance of PGM. The Group module plays a pivotal role in optimizing parameter allocation, thereby enhancing both the model's capacity and accuracy by efficiently utilizing available network resources. This module is particularly effective in balancing model size with performance, making it well-suited for scenarios with resource constraints. In contrast, the mask initialization module plays a crucial role in enhancing accuracy, especially in the context of subsequent tasks, as illustrated in Figure 2d. By offering an adaptive and structured starting point for mask optimization, it ensures efficient parameter usage for task-specific adaptations, minimizing task interference and reducing the risk of catastrophic forgetting. The consistent performance improvements observed across a variety of datasets highlight the complementary nature of these two components. Together, they enable PGM to achieve both high accuracy and efficient resource utilization, establishing a robust framework for incremental learning. This highlights the essential role of integrating both modules to effectively tackle the dual challenges of preserving task performance while managing model capacity.

**Different Group Size:** In the PGM framework, the size of each group plays a critical role in determining model perfor-

mance. Theoretically, as outlined in 3.2, larger group sizes tend to yield better results. However, empirical evidence reveals that beyond a certain value of $K$, performance tends to plateau, as shown in Figure 2c. Furthermore, it is not requisite for each group to select a fixed number of parameters. When $K = Mix$, the model allows for dynamic selection of parameter sizes across different group combinations, leading to enhanced performance. This flexibility contributes to more effective utilization of network resources, optimizing both task-specific adaptation and overall model efficiency.

**Adaptability to Diverse Training Paradigms:** Parameter isolation methods for mask selection can be broadly categorized into two primary approaches. The three-stage training paradigm, represented by PackNet (Mallya & Lazebnik, 2018), consists of sequential steps: training the entire network, selecting an optimal mask, and subsequently finetuning the selected sub-network. In contrast, the iterative training paradigm, as employed in WSN (Kang et al., 2022), integrates sub-network updates directly within the training process, enabling a more dynamic adaptation to task-specific requirements. As summarized in Table 3, PGM exhibits strong adaptability to both training paradigms. Nevertheless, the three-stage training approach exhibits limitations in knowledge reuse across tasks, resulting in increased overall parameter overhead and suboptimal resource efficiency. In contrast, the iterative training approach not only achieves superior accuracy but also significantly reduces capacity utilization, underscoring its effectiveness in optimizing parameter efficiency and computational resources.

*Table 3.* Comparison of training modes on the CIFAR-100 Superclass dataset in terms of ACC and CAP, where higher ACC and lower CAP indicate better performance and efficiency.

| Training Mode | ACC (%) ↑ | CAP (%) ↓ |
|---|---|---|
| Three-stage training | 62.42 | 76.26 |
| Iterative training | 62.56 | 65.97 |

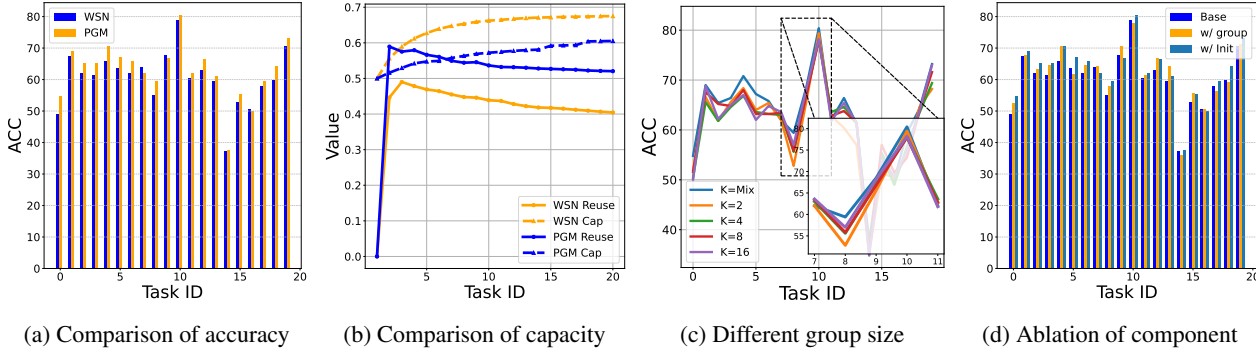

(a) Comparison of accuracy     (b) Comparison of capacity     (c) Different group size     (d) Ablation of component

*Figure 2.* Performance comparison across tasks. (a) Accuracy comparison across tasks, showing the performance differences in accuracy between WSN and our proposed method across different tasks. (b) Capacity comparison across tasks, illustrating the capacity used after training different tasks by WSN and our proposed method, as well as the overlap between the current and previous tasks. (c) Performance and compressed capacity variations with different $K$ values, showing that larger group sizes improve accuracy until performance plateaus. (d) Comparison of PGM with state-of-the-art methods, highlighting PGM's superior accuracy and efficiency.

## 4.4. Further Analysis

**Computational Efficiency:** The proposed method emphasizes a balance between computational efficiency and performance, offering practical advantages without significantly increasing time consumption. As depicted in Figure 4a, PGM consistently outperforms some methods such as Pack-Net, SupSup, and WSN in terms of convergence speed across various benchmark datasets. This result highlights PGM's ability to deliver superior performance with only a marginal increase in computational time. Furthermore, Figure 4a illustrates the time consumption for different values of $K$. Notably, even as $K$ increases, the time required grows only slightly, reflecting the method's capability to evaluate a larger number of parameter combinations without imposing a significant computational burden. These findings underline PGM's scalability and its ability to optimize parameters effectively while maintaining time efficiency, making it a robust and feasible solution for tasks.

**Parameter Dependency:** Figure 4b provides an in-depth analysis of parameter dependency patterns in convolutional layers. The results demonstrate that these dependencies predominantly follow a localized structure, with interactions primarily concentrated among adjacent parameters. Although broader dependencies occasionally manifest, the overall trend indicates that parameter interactions are largely confined to local neighborhoods. This observation suggests that, in most scenarios, global parameter dependencies may not require extensive consideration, thereby simplifying optimization processes and facilitating more efficient model design strategies. Moreover, the localized nature of dependencies implies that leveraging localized optimization techniques can be highly effective, reducing computational complexity without compromising performance. Figure 2c further corroborates this insight by illustrating that as the

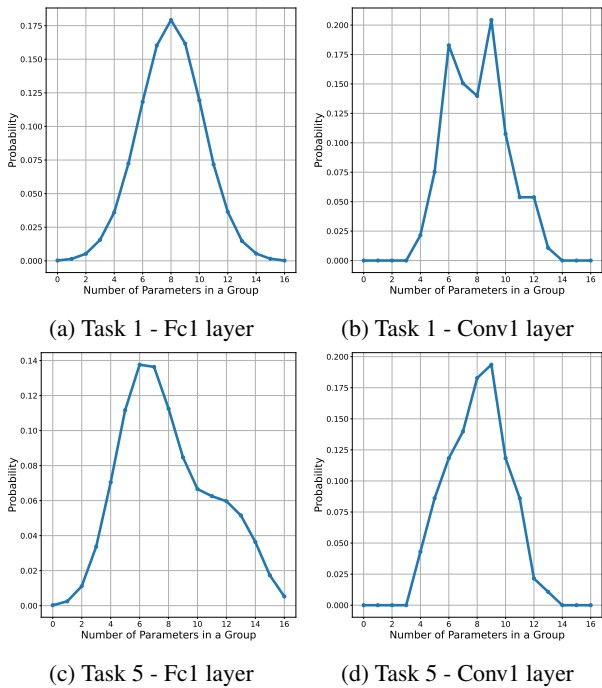

(a) Task 1 - Fc1 layer     (b) Task 1 - Conv1 layer

(c) Task 5 - Fc1 layer     (d) Task 5 - Conv1 layer

*Figure 3.* Visualization of MASK distributions across tasks in different layers of the CIFAR100Superclass dataset.

value of $K$ increases beyond a certain threshold, the model's performance stabilizes and exhibits diminishing sensitivity to additional increases. This trend underscores the effectiveness of optimizing local dependencies without the need for exhaustive exploration of global interactions, highlighting the scalability and efficiency of the proposed approach. Analysis of other layers is obtained in Appendix D.

**Task Differentiability:** Transforming task incremental learning (TIL) into a class incremental learning frame-

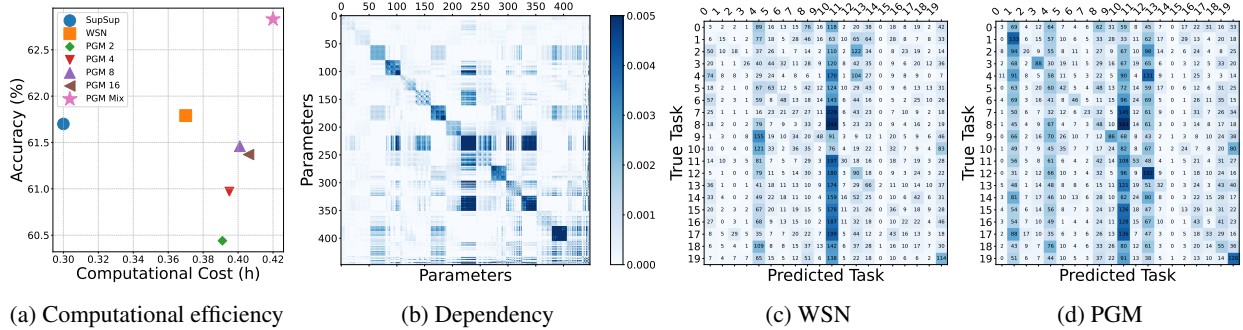

(a) Computational efficiency     (b) Dependency     (c) WSN     (d) PGM

*Figure 4.* Comprehensive Evaluation of PGM. (a) Computational efficiency comparison highlights PGM's scalability. (b) Dependency patterns within a layer showing localized correlations. (c, d) Confusion matrices illustrating task ID prediction using OOD scores for WSN and PGM, respectively, showing improved accuracy with an optimal sub-network.

work presents notable challenges, with effective Out-of-Distribution (OOD) detection (Liu et al., 2020; Xu & Yang, 2025) being crucial for accurate task identification during inference (Lin et al., 2024). Figures 4c and 4d illustrate the confusion matrices for task ID prediction, where OOD scores are computed by feeding input samples into each model and selecting the one with the highest energy score from class logits, as proposed in (Liu et al., 2020). Compared to WSN, PGM demonstrates a clearer diagonal structure in the confusion matrix, indicating improved task differentiation. This result suggests that PGM's sub-network selection strategy enhances task ID prediction accuracy by better capturing task-specific features and reducing task interference. These findings highlight the importance of optimizing sub-network selection to improve task differentiation in class incremental learning scenarios. More results can be found in the appendix E.

**Different Layer Parameter Distribution:** We visualize the parameter distribution across two tasks (Task1 and Task5) and different layers (Conv1 and Fc1) in the CIFAR100 Superclass dataset. The probabilities, including $p(j)$, which denotes the likelihood of selecting a specific number of parameters within a group, and the probability of selecting a particular combination pattern, are defined in the methods section. As shown in Figure 3, different layers exhibit distinct distribution patterns. Convolutional layers, which share parameters within local receptive fields, tend to have more parameters within each group, whereas the dense connectivity of fully connected layers results in fewer correlated parameters per group. Additionally, during incremental learning, convolutional layers exhibit higher parameter reuse compared to fully connected layers, leading to fewer parameters retained per group in later tasks. Additional visualizations of distributions for other tasks are provided in the appendix C.

## 5. Related Work

### 5.1. Incremental learning

To mitigate catastrophic forgetting (CF) in Incremental learning (IL), various methods have been proposed. Regularization-based approaches penalize changes to important parameters to preserve knowledge from previous tasks (Kirkpatrick et al., 2020; Lee et al., 2020), while rehearsal-based methods retain and replay key samples to support the learning of new tasks without forgetting prior ones (Yoon et al., 2022; Chaudhry et al., 2019). Distinct from these approaches, parameter isolation methods allocate distinct parameters to each task, ensuring that updates for new tasks do not interfere with those assigned to prior tasks, thereby effectively mitigating CF. For example, PackNet (Mallya & Lazebnik, 2018) and CLNP (Golkar et al., 2019) use $l_1$ regularization to identify and freeze important neurons while reinitializing unselected neurons for future tasks. Similarly, PathNet (Fernando et al., 2017) partitions each layer into multiple submodules and selects the optimal pathway for each task.However, these methods rely on handcrafted criteria to assess the importance of individual parameters, which often results in suboptimal allocations due to their inherent limitations. Alternatively, learnable importance measures have been introduced, enabling dynamic evaluation of parameter relevance during training and providing a more precise approach to parameter allocation. For instance, SPG (Konishi et al., 2023) promotes plasticity by employing parameter-level soft masking, selectively constraining updates to critical parameters based on their relevance to prior tasks. Similarly, WSN (Kang et al., 2022) learns task-specific subnetworks by jointly optimizing binary masks and reusing weights from prior tasks, enabling the identification of compact and effective subnetworks for each task.

## 6. Conclusion

In this paper, we improve parameter selection in IL by accounting for parameter dependencies. Unlike traditional methods that independently score parameters, PGM models the selection process as a probabilistic sampling process, grouping parameters and allowing for joint optimization. PGM utilizes Gumbel-Softmax to enable efficient mask selection. Our theoretical demonstrates that incorporating parameter dependencies enhances subnetwork selection, leading to more effective utilization of model capacity. The experimental results on datasets validate the superiority of PGM over existing IL techniques. Future work could explore further optimizations of mask selection processes, including the integration of more complex dependency structures or alternative optimization strategies.

## Acknowledgements

National Key RD Program of China (2022YFF0712100), NSFC (62276131), Natural Science Foundation of Jiangsu Province of China under Grant (BK20240081), the Fundamental Research Funds for the Central Universities (No.30922010317, No.30923011007), Key Laboratory of Target Cognition and Application Technology (2023-CXPT-LC-005).

## Impact Statement

This paper presents work whose goal is to advance the field of Machine Learning. There are many potential societal consequences of our work, none which we feel must be specifically highlighted here.

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

# A. Detailed Derivation of Error Combination Selection Method

To provide a more detailed derivation of the error differences between the Top-k selection and combination selection methods, we start with the fundamentals of model selection, constructing a mathematical model step by step to derive the error difference. By analyzing the sources of error, we can better understand how each method behaves in the selection process.

### Scoring Model and Error

For each parameter $\theta_i$ $(i = 1, 2, \ldots, N)$, there exists a true score $\text{TrueScore}(\theta_i)$. Due to the presence of scoring errors, the actual score of each parameter is represented as:

$$\text{Score}(\theta_i) = \text{TrueScore}(\theta_i) + \Delta_i$$

where $\Delta_i$ is the error associated with the true score, assumed to follow a Gaussian distribution $\Delta_i \sim \mathcal{N}(0, \sigma^2)$, with zero mean and variance $\sigma^2$.

### Error Analysis of Top-k Selection

In the Top-k method, we select parameters based on their independent scores. Let the set of selected parameters be $\mathcal{S}_{\text{top-k}}$, chosen by ranking the scores:

$$\mathcal{S}_{\text{top-k}} = \text{TopK}(\text{Score}(\theta_1), \text{Score}(\theta_2), \ldots, \text{Score}(\theta_N))$$

We then compute the set of incorrectly selected parameters $\mathcal{S}_{\text{wrong}}$, i.e., those that were chosen but do not belong to the ideal set $\mathcal{S}^*$. Due to the scoring errors, parameters $\theta_i^*$ and $\theta_j$ may be misselected. The condition for $\theta_j$ to be misselected is:

$$\text{Score}(\theta_j) > \text{Score}(\theta_i)$$

Substituting the scoring error model, we get:

$$\text{TS}(\theta_j) + \Delta_j > \text{TS}(\theta_i) + \Delta_i$$

Simplifying:

$$\Delta_j - \Delta_i > \text{TS}(\theta_i) - \text{TS}(\theta_j)$$

Assuming $\Delta_j - \Delta_i \sim \mathcal{N}(0, 2\sigma^2)$, the probability of error selection $P_{\text{wrong}}(i, j)$ is:

$$P_{\text{wrong}}(i, j) = \Pr\left(\Delta_j - \Delta_i > \text{TS}(\theta_i) - \text{TS}(\theta_j)\right)$$

This probability can be expressed using the cumulative distribution function of the standard normal distribution:

$$P_{\text{wrong}}(i, j) = \Phi\left(-\frac{\text{TS}(\theta_i) - \text{TS}(\theta_j)}{\sqrt{2}\sigma}\right)$$

### Error Analysis of Combination Selection

In the combination selection method, we evaluate all possible combinations of $k$ parameters. Let the selected parameter combination be $\mathcal{C}$, and its score is:

$$\mathcal{S}_{\text{combo}} = \arg\max_{\mathcal{C} \in C_N^k} \sum_{i \in \mathcal{C}} \text{Score}(\theta_i)$$

In combination selection, we consider the errors of all $k$ parameters. For each combination, we have:

$$\sum_{i \in \mathcal{C}} \text{Score}(\theta_i) = \sum_{i \in \mathcal{C}} (\text{TS}(\theta_i) + \Delta_i)$$

This combined score follows a Gaussian distribution:

$$\sum_{i \in \mathcal{C}} \text{Score}(\theta_i) \sim \mathcal{N}\left(\sum_{i \in \mathcal{C}} \text{TS}(\theta_i), k\sigma^2\right)$$

Thus, the probability of error selection for the combination is:

$$P_{\text{wrong}}(\mathcal{C}) = \Phi\left(-\frac{\sum_{i \in \mathcal{S}^*} \text{TS}(\theta_i) - \sum_{i \in \mathcal{C}} \text{TS}(\theta_i)}{\sqrt{k}\sigma}\right)$$

This indicates that, due to the combination of $k$ parameters, the error variance is reduced by a factor of $\sqrt{k}$, leading to a smaller overall error.

## B. Implementation Details

For a fair comparison, we strictly adhere to the experimental settings outlined in (Kang et al., 2022), including the use of identical backbone networks for the corresponding datasets. Specifically, we employ a modified version of AlexNet, as proposed by (Serrà et al., 2018), for the Split CIFAR-100 dataset, and a modified LeNet architecture, as described by (Saha et al., 2021), for the CIFAR-100 Superclass dataset. For Split TinyImageNet, we adopt the backbone introduced by (Kang et al., 2022), which comprises a 4-layer convolutional structure followed by 3 fully connected layers. In line with (Kang et al., 2022), we use the Adam optimizer with a momentum of 0.9 for initial model training. Each task is trained for 50 epochs on CIFAR-100 and 40 epochs on Split TinyImageNet. All experiments are implemented using PyTorch on a system equipped with four NVIDIA 4090 GPUs.

## C. Different Layer and Parameter Distribution

Here, we visualize all layers of LeNet on the CIFAR Superclass across different tasks. The results confirm our previous observation: convolutional layers exhibit greater parameter sharing compared to other layers.

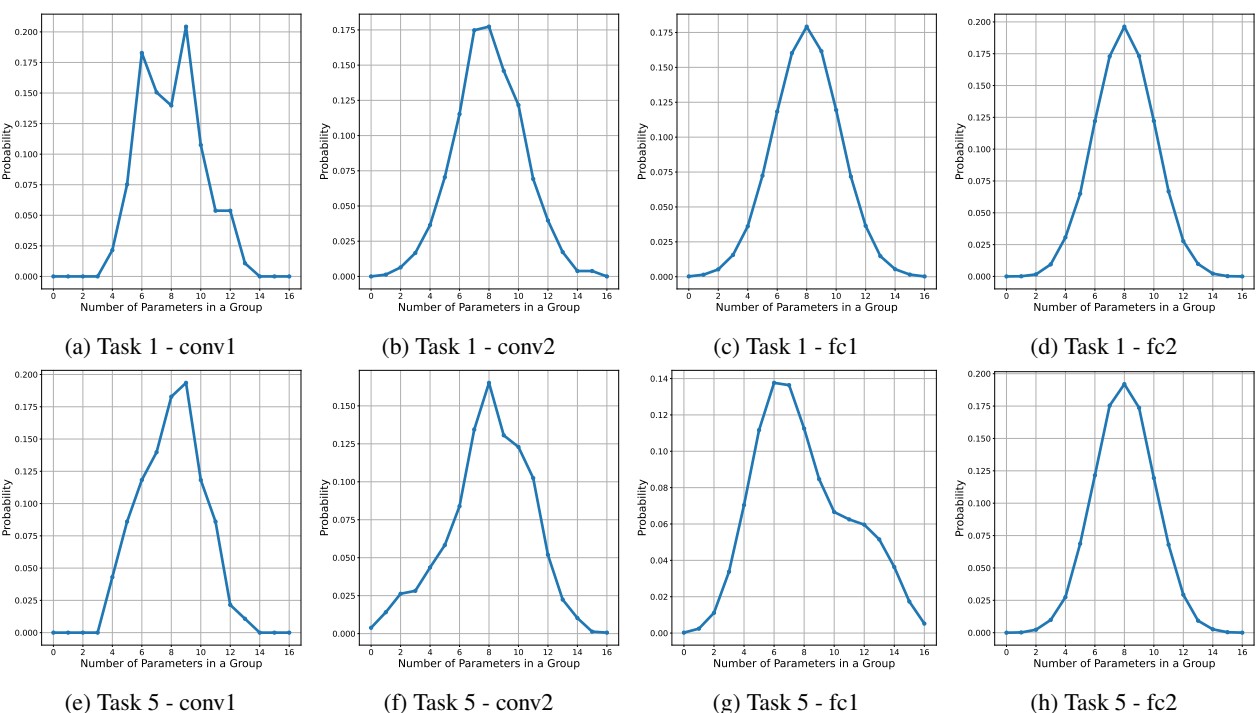

*Figure 5.* Weight distributions across multiple tasks

## D. Parameter Dependency Across Different Layers

This section presents the parameter dependencies in different layers. As illustrated in the figure below, the convolutional layer exhibits stronger parameter correlations compared to the fully connected layer, where dependencies are more sparse and dispersed. This observation highlights the structured nature of convolutional filters, which inherently capture local patterns and share spatial correlations, whereas fully connected layers tend to have less structured parameter interactions.

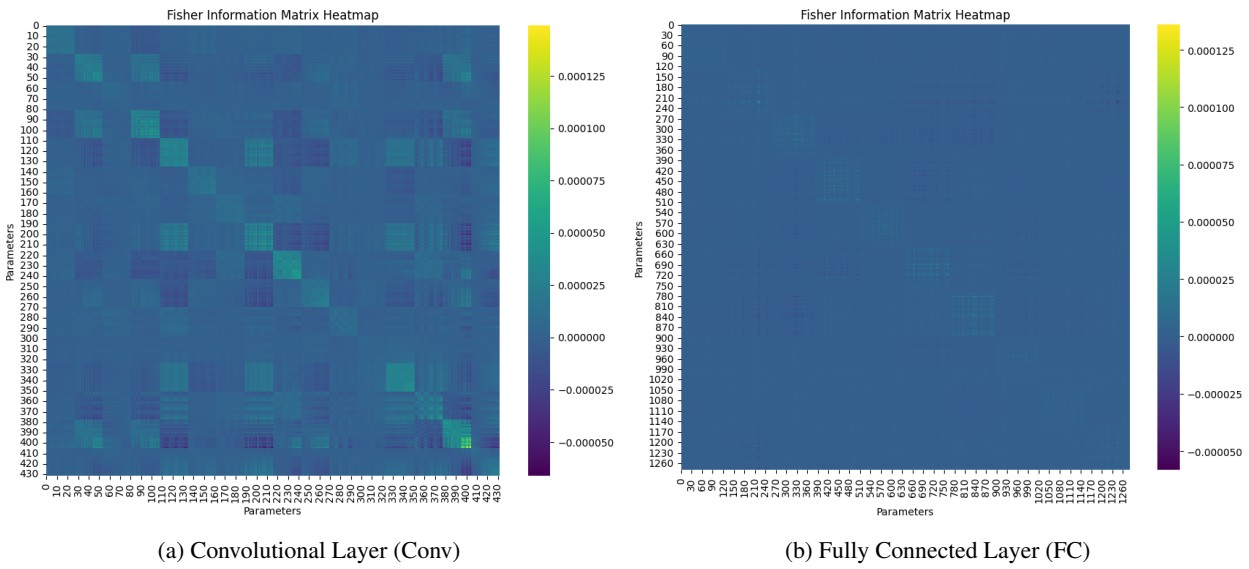

(a) Convolutional Layer (Conv)                    (b) Fully Connected Layer (FC)

*Figure 6.* Visualization of parameter dependency across different layers. The left subfigure illustrates dependencies within the convolutional layer, while the right subfigure shows those in the fully connected layer.

## E. Extend to Class-Incremental Learning

The proposed method is developed under the task-incremental learning setting, where parameter isolation methods prevents forgetting via task-specific subnetworks. However, as noted in (Kim et al., 2022), forgetting may re-emerge in the more challenging class-incremental learning (CIL) setting due to the absence of task identity during inference. To address this, (Kim et al., 2022) introduces OOD detection for implicit task inference, and (Lin et al., 2024) further enhances it with rehearsal samples. Following the protocol in (Lin et al., 2024), we replace their mask selection strategy with ours, and compare against parameter-isolation baselines without additional memory. As shown in Table 4, our method achieves competitive performance in the CIL setting.

*Table 4.* CIL performance comparison on CIFAR100-10 using DeiT as the backbone. "Last" denotes the accuracy after learning the final task, and "AIA" denotes the average incremental accuracy.

| Method | WSN | | PGM | |
|---|---|---|---|---|
| | Last ↑ | AIA ↑ | Last ↑ | AIA ↑ |
| Energy | 62.21 | 75.19 | **64.43** | **77.06** |
| TPL | 67.89 | 80.93 | **69.55** | **81.78** |

## F. Generalization to Non-Vision tasks

To ensure fairness and comparability, we adopt the same task setup as in (Kang et al., 2022; Hu et al., 2024), which are widely used as baselines for evaluating incremental learning performance. To further assess the generalization capability of our method beyond the vision domain, we extend it to an audio classification task using the KineticsSounds dataset (Arandjelovic & Zisserman, 2017). The dataset is divided into five incremental tasks, denoted as KS-5. As shown in Table 1, PGM outperforms WSN in both accuracy and parameter capacity when using the ResNet18 architecture.

Table 5. Performance evaluation on the KS dataset.

| Method | WSN | | PGM | |
|---|---|---|---|---|
| | Acc ↑ | CAP ↓ | Acc ↑ | CAP ↓ |
| KS-5 | 69.43 | 76.44 | **70.44** | **57.41** |

## G. Generalization across Model Architectures

To further assess the generalization capability of PGM across different architectures, we evaluate it on both DeiT and ResNet18 following the settings in (Lin et al., 2024; Kang et al., 2022), and compare it with parameter-isolation methods that incur no additional storage cost and exhibit no forgetting. As shown in Table 6, PGM consistently demonstrates robust performance across all evaluated configurations. Notably, the greater parameter reduction observed on ResNet architectures can be attributed to the larger number of convolutional layers, where modeling parameter dependency tends to be more effective. In contrast, Transformer-based architectures contain more linear layers, where parameter dependency are inherently weaker, leading to comparatively smaller gains.

Table 6. Comparative performance evaluation across different model architectures on CIFAR100-10.

| Method | WSN | | PGM | |
|---|---|---|---|---|
| | Acc ↑ | CAP ↓ | Acc ↑ | CAP ↓ |
| ResNet18 | 73.51 | 90.66 | **75.37** | **71.59** |
| DeiT | 93.78 | 69.46 | **94.21** | **60.65** |

