# OpenReview forum: "Probabilistic Group Mask Guided Discrete Optimization for Incremental Learning"
_ICML.cc/2025/Conference — ICML 2025 poster_

### Official Review · Reviewer_km4w · 2025-03-07

**Overall Recommendation:** 4

**Summary:**

This paper concerns the parameter-isolation methods in incremental learning. However, existing approaches often disregard parameter dependencies, resulting in an over-reliance on newly allocated parameters. To address this issue, this paper proposes Probabilistic Group Mask selection (PGM), a group-wise approach that captures parameter dependencies by exploring candidate masks within each group. Specifically, PGM partitions parameters into groups with multiple candidate masks, assigning probabilities to these masks and leveraging Gumbel-Softmax for differentiable sampling, enabling efficient optimization of the discrete mask selection process. The theoretical analysis demonstrates that incorporating parameter dependencies enhances sub-network selection. Experiments conducted on standard benchmarks confirm its superior effectiveness compared to existing IL approaches.

## update after rebuttal

**Claims And Evidence:**

The claims made in the submission are supported by clear and convincing evidence.

**Essential References Not Discussed:**

No.

**Experimental Designs Or Analyses:**

Yes. The soundness and validity of experimental designs or analyses are make sense.

**Methods And Evaluation Criteria:**

Yes. It makes sense.

**Other Comments Or Suggestions:**

1. The writing is clear, but the appendix could better explain implementation details (e.g., hyperparameters for Gumbel-Softmax).

**Other Strengths And Weaknesses:**

Strengths
1. Innovative Approach: This paper proposes Probabilistic Group Mask selection (PGM), a group-wise approach that captures parameter dependencies by exploring candidate masks within each group. The group-wise probabilistic masking strategy effectively captures parameter dependencies, a novel contribution to parameter-isolation methods.

2. Theoretical Grounding: The theoretical analysis demonstrates that incorporating parameter dependencies enhances sub-network selection. The formal analysis of parameter reuse and error reduction via dependency modeling strengthens the method’s motivation.

3. Empirical Results: PGM outperforms baselines like WSN across metrics, particularly in reducing parameter capacity while maintaining accuracy. Meanwhile，the further analysis reveal the effectiveness of each module.

4. Good Analysis: The ablation studies, computational efficiency tests, and visualizations (e.g., dependency patterns, mask distributions) provide valuable insights.

Weaknesses:

1. Limited Task Diversity: Experiments focus on image classification; testing on non-vision tasks (e.g., NLP) would strengthen generalizability claims.

2. Group Size Sensitivity: While larger groups improve performance, the plateau effect at higher K is not thoroughly analyzed (e.g., computational trade-offs).

**Questions For Authors:**

1. Could the method be extended to class-incremental learning (CIL) without task IDs?

2. What is the overhead of maintaining group-wise masks for large-scale models (e.g., Transformers)?

**Relation To Broader Scientific Literature:**

This paper is related to class-incremental learning, which is widely researched.

**Theoretical Claims:**

Yes.

---

> ### Author Rebuttal · Authors · 2025-03-31
>
> Thank you for your valuable comments.
>
> **Generalization to Non-Vision tasks:** To ensure fairness and comparability, we adopt the same task setup as in [1,2], which are widely used as baselines for evaluating incremental learning performance. To further assess the generalization capability of our method beyond the vision domain, we extend it to an audio classification task using the KineticsSounds dataset [3]. The dataset is divided into five incremental tasks, denoted as KS-5. As shown in Table 1, PGM outperforms WSN in both accuracy and parameter capacity when using the ResNet18 architecture.
>
> **Table 1.** Performance evaluation on the KS dataset.
> |Architecture|Method|KS-5||
> |-|-|-|-|
> |||Acc↑|CAP↓|
> |ResNet18|WSN|69.43|76.44|
> ||PGM|**70.44**|**57.41**|
>
> **Analysis of the Plateau Effect at Larger Group Sizes:** While the theoretical search space for mask combinations grows exponentially with group size $K$, our Gumbel-Softmax sampling reduces the practical complexity to linear time (i.e., $\mathcal{O}(K)$). As shown in Figure 4a of the original paper, while computational overhead increases only marginally with larger \(K\), the average accuracy begins to plateau beyond a certain threshold. More detailed hyperparameter settings will be given in the final version.
>
> **Extended to Class Incremental Learning**: Extending task-incremental learning to class-incremental learning requires accurate task identity recognition. Prior works [4,5] have shown that accurate task-ID prediction in the CIL setting relies on strong out-of-distribution (OOD) detection capabilities. To this end, we integrated the Energy-based OOD detection method [6] into our framework. As shown in Table 2, PGM achieves higher accuracy than WSN on both the Last and AIA metrics, suggesting that the subnetworks selected by PGM demonstrate stronger OOD detection capability.
>
> **Table 2.** CIL performance comparison using DeiT architecture. AIA is the average incremental ACC. Last is the ACC after learning the final task.
> |Architecture|Method|CIFAR100-10||
> |-|-|-|-|
> |||Last↑|AIA↑|
> |Deit|WSN+Energy|62.21|75.19|
> ||PGM+Energy|**64.43**|**77.06**|
>
> **Training Time on Large Scale Models**: To ensure fairness and comparability, we adopt the same model architectures as in [1,2], which are widely recognized as baselines for incremental learning. To further assess the training time of PGM across different architectures, we follow the settings in [5,7] and evaluate it on both ResNet18 and DeiT, comparing against parameter-isolation methods that require no additional storage and exhibit no forgetting. As shown in Table 3, the joint evaluation and optimization of grouped parameters introduce noticeable computational overhead as model size increases. While this results in longer training time, it helps reduce the risk of suboptimal parameter selection by explicitly modeling parameter dependency.
>
> **Table 3.** Performance evaluation across different model architectures.
> |Architecture|Method|CIFAR100-10|||
> |-|-|-|-|-|
> |||Acc↑|CAP↓|Time(h)↓|
> |ResNet18|WSN|73.51|90.66|0.32|
> ||PGM|**75.37**|**71.59**|0.43|
> |DeiT|WSN|93.78|69.46|0.55|
> ||PGM|**94.21**|**60.65**|0.76|
>
> **References**:
> [1]. Haeyong Kang, et.al. Forget-free continual learning with winning subnetworks. *ICML*, 2022.\
> [2]. Yusong Hu, et.al. Task-aware Orthogonal Sparse Network for Exploring Shared Knowledge in Continual Learning. *ICML*, 2024.\
> [3]. Relja Arandjelovic, et.al. Look, listen and learn. *ICCV*, 2017.\
> [4]. Gyuhak Kim, et.al. A Theoretical Study on Solving Continual Learning. *NeurIPS*, 2022\
> [5]. Haowei Lin, et.al. Class incremental learning via likelihood ratio based task prediction. *ICLR*, 2024.\
> [6]. Weitang Liu, et.al. Energy-based out-of-distribution detection. *NeurIPS*, 2020.\
> [7]. Md. Sazzad Hossain, et.al. Rethinking Task-Incremental Learning Baselines. *ICPR*, 2022.

---

### Official Review · Reviewer_uCBE · 2025-03-09

**Overall Recommendation:** 4

**Summary:**

The authors propose PGM, a dependency-aware parameter-isolation framework for IL that optimizes task-specific subnetworks via probabilistic group masking. By grouping parameters and sampling masks with Gumbel-Softmax, PGM improves parameter reuse and reduces capacity overhead. Theoretical proofs link dependency modeling to improved sub-network selection. Extensive experiments are conducted to show the SOTA results on standard benchmarks. The work includes ablation studies, efficiency analysis, and visualizations of parameter distributions. The author has thoroughly substantiated the paper’s arguments from multiple perspectives, including methodology, theoretical analysis, and experiments, thereby strongly supporting the proposed viewpoints.

**Claims And Evidence:**

Yes. The main claims in this paper are supported from methodology, theoretical analysis and experiments.

**Essential References Not Discussed:**

No

**Experimental Designs Or Analyses:**

Main experimental results demonstrate the superiority of the PGM. And the authors provide the ablation study, Computational Efficiency, Parameter Dependency, Different Layer Parameter Distribution to support the claims of the paper.

**Methods And Evaluation Criteria:**

Yes. The PGM is well-designed and effectively address the identified challenges in Incremental learning.

**Other Comments Or Suggestions:**

1. The dependency analysis (Fig. 5b) is insightful but could be expanded—e.g., how do dependencies vary across layers/tasks?
2. The impact of group size K on training time (Fig. 6a) is underdiscussed. A complexity analysis would help.

**Other Strengths And Weaknesses:**

Strengths
1. Practical Impact: The efficient parameter reuse mechanism in PGM significantly reduces memory and computational overhead, making it particularly well-suited for deployment in resource-constrained environments.
2. Differentiable Optimization: The Gumbel-Softmax reparameterization elegantly handles discrete mask selection, enabling end-to-end training. The learning algorithm is well-designed and novel.
3. Task-Informed Initialization: Leveraging prior task masks for initialization is an innovative strategy that enhances transferability by effectively retaining relevant knowledge from previous tasks.
4. The paper rigorously validates its claims from multiple perspectives, including methodological soundness, theoretical justification, comprehensive ablation studies, efficiency analysis, and insightful visualizations. This multifaceted evaluation ensures a thorough understanding of the proposed approach, reinforcing its reliability and effectiveness.
Weaknesses
1. Long-Task Scalability: The experiments are limited to sequences of 5–10 tasks, which may not fully capture the scalability of the approach. Extending the evaluation to longer task sequences would provide deeper insights into its long-term performance, stability, and adaptability in more complex scenarios.
2. How does PGM ensure stability when dependencies conflict across tasks (e.g., Task 1’s critical parameters are Task 2’s noise
3. For the TinyImageNet results, why is the ACC improvement over SPG smaller compared to CIFAR-100?

**Questions For Authors:**

Could the method integrate rehearsal or generative replay to further reduce forgetting?

**Relation To Broader Scientific Literature:**

The key contributions of the paper are related to the incremental learning.

**Theoretical Claims:**

Yes. The proposed theory, i.e., Error Reduction via Dependency, is rigorously proven to be correct and strongly supports the viewpoints of the paper.

---

> ### Author Rebuttal · Authors · 2025-03-31
>
> Thank you for your valuable comments.
>
> **Extending to Longer Task Sequences:** To ensure fairness and comparability, we adopt the same task configurations as in [1,2] (i.e., 10, 20, and 40 tasks), which are widely recognized as baselines for evaluating incremental learning performance. To further assess performance under longer task sequences, we extend the evaluation on CIFAR-100 by increasing the number of tasks to 10, 20, and 50 under the DeiT architecture [4]. As shown in Table 1, PGM consistently achieves strong performance across all settings and demonstrates better scalability than WSN, particularly with respect to CAP.
>
> **Table 1.** Performance evaluation on long task sequences using DeiT architecture.
> |Architecture|Method| CIFAR100-10||CIFAR100-20||CIFAR100-50||
> |-|-|-|-|-|-|-|-|
> |||Acc↑|CAP↓|Acc↑|CAP↓|Acc↑|CAP↓|
> |DeiT|WSN|93.78|69.46|97.68|72.84|98.40|76.25|
> ||PGM|**94.21**|**60.65**|**98.22**|**65.33**|**98.91**|**70.64**|
>
> **Stability under Conflicting Dependency across Tasks and Layers:** Parameter dependency are modeled at the intra-layer levels, reflecting the fact that the importance of a parameter can be influenced by other parameters within the same task. To facilitate knowledge transfer while reducing task interference, we employ a similarity-based mask initialization strategy that modulates the influence of prior tasks according to their similarity to the current task. Greater task discrepancy results in lower similarity scores, which suppress the transfer of conflicting knowledge. While the current framework focuses on modeling intra-layer dependency, incorporating cross-layer dependency may further improve performance and generalization. This remains a promising direction for future work, and we plan to explore it in subsequent research.
>
> **Different Performance Gains on TinyImageNet Compared to CIFAR-100:** The difference in ACC improvement can be attributed to variations in dataset characteristics and model configurations. TinyImageNet uses images with a resolution of 64×64, includes 40 tasks, and is trained with a network composed of four convolutional layers and three fully connected layers. In contrast, CIFAR-100 uses images with a resolution of 32×32 and includes 10 and 20 tasks, with AlexNet used for the 10-task setting and LeNet for the 20-task setting. These differences result in performance variation across the two benchmarks.
>
> **Impact of Group Size K on Complexity and Training Time:** While the theoretical search space for mask combinations grows exponentially with group size $K$, our Gumbel-Softmax sampling reduces the practical complexity to linear time (i.e., $\mathcal{O}(K)$). This is achieved by directly learning parameter activation distributions, thereby avoiding exhaustive combinatorial search. As shown in Figure 4a of the original paper, performance gains tend to plateau beyond K = 8, suggesting that further increasing the group size yields limited benefit while introducing additional computational overhead.
>
> **Extending to Class-Incremental Learning with Rehearsal Sample:** The proposed method is developed under the task-incremental learning setting, where parameter isolation methods prevents forgetting via task-specific subnetworks. However, as noted in [3], forgetting may re-emerge in the more challenging class-incremental learning (CIL) setting due to the absence of task identity during inference. To address this, [3] introduces OOD detection for implicit task inference, and [4] further enhances it with rehearsal samples. Following the protocol in [4], we replace their mask selection strategy with ours, and compare against parameter-isolation baselines without additional memory. As shown in Table 2, our method achieves competitive performance in the CIL setting.
>
> **Table 2.** CIL performance comparison using DeiT architecture. AIA is the average incremental ACC. Last is the ACC after learning the final task.
> |Architecture|Method|CIFAR100-10||
> |-|-|-|-|
> |||Last↑|AIA↑|
> |Deit|WSN+TPL|67.89|80.93|
> ||PGM+TPL|**69.55**|**81.78**|
>
> **References**:
> [1]. Haeyong Kang, et.al. Forget-free continual learning with winning subnetworks. *ICML*, 2022.\
> [2]. Yusong Hu, et.al. Task-aware Orthogonal Sparse Network for Exploring Shared Knowledge in Continual Learning. *ICML*, 2024.\
> [3]. Gyuhak Kim, et.al. A Theoretical Study on Solving Continual Learning. *NeurIPS*, 2022\
> [4]. Haowei Lin, et.al. Class incremental learning via likelihood ratio based task prediction. *ICLR*, 2024.

---

### Official Review · Reviewer_uc5U · 2025-03-12

**Overall Recommendation:** 4

**Summary:**

This paper introduces Probabilistic Group Mask (PGM), a novel parameter-isolation method for incremental learning (IL) that addresses catastrophic forgetting by incorporating parameter dependencies during sub-network selection. PGM groups parameters, assigns probabilistic masks via Gumbel-Softmax Sampling, and dynamically initializes masks based on prior task similarities. Theorem 3.2 states that grouping parameters minimizes selection error by utilizing dependencies. Comprehensive experimental results on Split CIFAR100, CIFAR-100 Superclass, and Split TinyImagenet demonstrate PGM’s superior accuracy and parameter efficiency compared to existing methods like WSN [ICML’22], SPG [ICML’23], RP2F [ACMMM’24], with reduced reliance on new parameters and improved task differentiation.

**Claims And Evidence:**

Yes. Claim “incorporating parameter dependencies” is supported by Theorem 3.2 and empirical results. Claim “Group-wise mask selection” is supported by derivation of differentiable sampling and ablation study.

**Essential References Not Discussed:**

The references have been thoroughly discussed. All key references have been incorporated, particularly those concerning the most recent advancements in incremental learning.

**Experimental Designs Or Analyses:**

The author presents the extensive experiments including main comparison with recent state-of-the-art baselines of incremental learning, ablation study for effectiveness of key components, influence of group size, adaptability to diverse training paradigms.

**Methods And Evaluation Criteria:**

Yes. The proposed PGM effectively addresses the challenges, i.e., catastrophic forgetting by incorporating parameter dependencies, in incremental learning.

**Other Comments Or Suggestions:**

See weaknesses.

**Other Strengths And Weaknesses:**

Strengths:
1. Novel integration of probabilistic group masking and parameter dependencies. PGM introduces a group-wise strategy that explicitly models parameter interactions, a significant departure from existing methods that treat parameters independently.
2. Strong empirical results across metrics and datasets. The method achieves state-of-the-art results on Split CIFAR-100, CIFAR-100 Superclass, and Split TinyImageNet, outperforming parameter-isolation baselines like WSN and regularization methods like GPM. Furthermore, the authors analyze the computational cost, parameter dependency, task differentiability.
3. Effective use of Gumbel-Softmax for differentiable optimization. The reparameterization of discrete mask selection via Gumbel-Softmax (Eq. 4) enables end-to-end training while preserving task-specific adaptability.
4. Comprehensive ablation study validating design choices. The paper rigorously validates individual contributions of the Group and Mask Initialization modules (Table 2), showing that their combination drives performance gains.
Weaknesses:
1. Limited scalability analysis for large models: Experiments focus on smaller architectures (e.g., AlexNet, LeNet), leaving scalability to modern networks (e.g., ResNet, ViTs) unaddressed.
2. Superficial treatment of dependency patterns: Although Figure 6 visualizes localized dependencies in convolutional layers, the work does not quantify their impact (e.g., correlation strength) or explore adaptive grouping strategies.

**Questions For Authors:**

How does PGM handle scenarios where parameter dependencies are non-local (e.g., in transformers)?
Why is CAP defined with Huffman encoding (Sec. 4.1), and how does this affect interpretation?
Furthermore, please refer to weakness for more questions.

**Relation To Broader Scientific Literature:**

Yes. This paper focuses on core issue, i.e., catastrophic forgetting by incorporating parameter dependencies, in incremental learning.

**Theoretical Claims:**

Yes. The theory proposed in the paper is sound. Theorem 3.2 establishes a mathematical basis, demonstrating how group-wise selection reduces error rates through variance reduction. Although based on a simplified assumption (Gaussian errors), this analysis is consistent with empirical findings (Fig. 2c) and highlights the group size K as an adjustable parameter balancing accuracy and complexity.

---

> ### Author Rebuttal · Authors · 2025-03-31
>
> Thank you for your valuable comments.
>
> **Scalability to Modern Architectures:** To ensure the fairness and comparability of results, we adopt the same model architectures as in [1,2], which are widely recognized as baselines for evaluating incremental learning performance. To further assess the generalization capability of PGM across different architectures, we evaluate it on both DeiT and ResNet18 following the settings in [3,4], and compare it with parameter-isolation methods that incur no additional storage cost and exhibit no forgetting. As shown in Table 1, PGM consistently demonstrates robust performance across all evaluated configurations. Notably, the greater parameter reduction observed on ResNet architectures can be attributed to the larger number of convolutional layers, where modeling parameter dependency tends to be more effective. In contrast, Transformer-based architectures contain more linear layers, where parameter dependency are inherently weaker, leading to comparatively smaller gains.
>
> **Table 1.** Performance evaluation across different model architectures.
> |Architecture|Method|CIFAR100-10||
> |-|-|-|-|
> |||Acc↑|CAP↓|
> |ResNet18|WSN|73.51|90.66|
> ||PGM|**75.37**|**71.59**|
> |DeiT|WSN|93.78|69.46|
> ||PGM|**94.21**|**60.65**|
>
> **On Correlation Quantification and Adaptive Grouping Strategies:** Figure 6 of the original paper shows that parameter interactions are predominantly local, with each parameter mainly influenced by its immediate neighbors, while the impact of distant parameters is considerably weaker. This observation supports the assumption that grouped parameter blocks can be treated as approximately independent, suggesting that a simplified grouping strategy is sufficient to capture essential dependencies without significant loss in modeling capacity. Incorporating correlation-aware analysis and adaptive grouping mechanisms remains a promising direction for future research.
>
> **Huffman Encoding in CAP Calculation:** To ensure fair comparison with prior work [1], we adopt Huffman encoding for consistency. The binary nature of the mask (i.e., 0/1 values) aligns well with Huffman's optimal prefix coding scheme, allowing for efficient compression and significant reduction in mask storage.
>
> **References**:
> [1]. Haeyong Kang, et.al. Forget-free continual learning with winning subnetworks. *ICML*, 2022.\
> [2]. Yusong Hu, et.al. Task-aware Orthogonal Sparse Network for Exploring Shared Knowledge in Continual Learning. *ICML*, 2024.\
> [3]. Haowei Lin, et.al. Class incremental learning via likelihood ratio based task prediction. *ICLR*, 2024.\
> [4]. Md. Sazzad Hossain, et.al. Rethinking Task-Incremental Learning Baselines. *ICPR*, 2022.

---

### Official Review · Reviewer_MRKd · 2025-03-18

**Overall Recommendation:** 3

**Summary:**

The paper introduces Probabilistic Group Mask (PGM) for incremental learning (IL), addressing catastrophic forgetting by modeling parameter dependencies during sub-network selection. Unlike prior methods that independently score parameters, PGM partitions parameters into groups and optimizes task-specific masks via probabilistic sampling within each group, using Gumbel-Softmax to enable differentiable optimization. Theoretical analysis shows grouping reduces selection errors by leveraging local parameter interactions, while task-aware initialization enhances reuse by aligning mask probabilities with prior task similarities. Experiments on Split CIFAR-100, CIFAR-100 Superclass, and Split TinyImageNet demonstrate PGM's superiority, achieving state-of-the-art accuracy and near-zero forgetting, underscoring its effectiveness in balancing stability and plasticity for scalable IL.

**Claims And Evidence:**

The claims made in the submission are well-supported by both theoretical analysis and empirical results.

**Essential References Not Discussed:**

N/A

**Experimental Designs Or Analyses:**

The experimental evaluation is comprehensive and rigorous. The authors test PGM on three standard benchmark datasets: Split CIFAR-100, CIFAR-100 Superclass, and Split TinyImageNet. They compare PGM against multiple state-of-the-art methods, including parameter isolation approaches (PackNet, SupSup, WSN), parameter regularization techniques (La-MAML, GPM, FS-DGPM), and other relevant baselines. The evaluation metrics include ACC (average classification performance), CAP (parameter capacity usage), and BWT (backward transfer). The results consistently show that PGM outperforms existing methods across all metrics.

**Methods And Evaluation Criteria:**

The paper presents a novel theoretical framework for understanding parameter selection in incremental learning. The authors introduce the concept of parameter reuse with dependency (Definition 3.1) and demonstrate through Theorem 3.2 that group-wise selection reduces evaluation errors by capturing local parameter interactions. This theoretical foundation justifies the group-wise approach and provides insight into why considering parameter dependencies leads to better sub-network selection.

**Other Comments Or Suggestions:**

No

**Other Strengths And Weaknesses:**

Advantages:
1. Theoretical Contributions: The paper provides a novel theoretical framework for understanding parameter selection in incremental learning, which is a significant contribution to the field.
2. Practical Efficiency: The method balances computational efficiency with performance, making it feasible for real-world applications.
3. Scalability: The group-wise approach allows the method to scale effectively with increasing numbers of tasks and parameters.

Disadvantages:
1. Implementation Complexity: The probabilistic sampling and Gumbel-Softmax reparameterization may increase implementation complexity compared to simpler mask selection methods.
2. Hyperparameter Sensitivity: Performance may be sensitive to the choice of group size (K) and other hyperparameters, requiring careful tuning for optimal results.
3. Limited Generalization Analysis: While the method performs well on the tested datasets, the paper could benefit from more extensive analysis of its generalization capabilities across different types of tasks and model architectures.

**Questions For Authors:**

None

**Relation To Broader Scientific Literature:**

The paper situates itself within the broader incremental learning literature, building upon and advancing parameter isolation methods. It acknowledges previous work on mask-based approaches while highlighting the novel contribution of incorporating parameter dependencies through group-wise selection. The approach aligns with recent trends in efficient machine learning that seek to optimize model capacity while maintaining performance.

**Theoretical Claims:**

Yes

---

> ### Author Rebuttal · Authors · 2025-03-31
>
> Thank you for your valuable comments.
>
> **Training Overhead and Parameter Sensitivity:** While probabilistic sampling and Gumbel-Softmax reparameterization may increase implementation complexity, this design enables differentiable and learnable mask selection, which is essential for effective parameter grouping. Larger group sizes $K$ can incur longer training time, and we observe that the performance improvement tends to plateau as $K$ increases (see Figure 4a of the original paper). Regarding hyperparameter sensitivity, the optimal value of $K$ may vary across datasets due to differences in task difficulty and parameter dependency. More detailed hyperparameter settings will be given in the final version.
>
> **Generalization across Model Architectures.**: To ensure the fairness and comparability of results, we adopt the same model architectures as in [1,2], which are widely recognized as baselines for evaluating incremental learning performance. To further assess the generalization capability of PGM across different architectures, we evaluate it on both DeiT and ResNet18 following the settings in [3,4], and compare it with parameter-isolation methods that incur no additional storage cost and exhibit no forgetting. As shown in Table 1, PGM consistently demonstrates robust performance across all evaluated configurations. Notably, the greater parameter reduction observed on ResNet architectures can be attributed to the larger number of convolutional layers, where modeling parameter dependency tends to be more effective. In contrast, Transformer-based architectures contain more linear layers, where parameter dependency are inherently weaker, leading to comparatively smaller gains.
>
> **Table 1.** Comparative performance evaluation across different model architectures.
> |Architecture|Method|CIFAR100-10||
> |-|-|-|-|
> |||Acc↑|CAP↓|
> |ResNet18|WSN|73.51|90.66|
> ||PGM|**75.37**|**71.59**|
> |DeiT|WSN|93.78|69.46|
> ||PGM|**94.21**|**60.65**|
>
> **Generalization across Different Task Types:** To further evaluate the generalization capability of our method beyond the vision domain, we extend it to an audio classification task using the KineticsSounds dataset [5]. The dataset is partitioned into five incremental tasks, referred to as KS-5. As shown in Table 2, PGM outperforms WSN in both accuracy and parameter capacity when using the ResNet18 architecture.
>
> **Table 2.** Comparative performance evaluation on the KS dataset.
> |Architecture|Method|KS-5||
> |-|-|-|-|
> |||Acc↑|CAP↓|
> |ResNet18|WSN|69.43|76.44|
> ||PGM|**70.44**|**57.41**|
>
> **References**:
> [1]. Haeyong Kang, et.al. Forget-free continual learning with winning subnetworks. *ICML*, 2022.\
> [2]. Yusong Hu, et.al. Task-aware Orthogonal Sparse Network for Exploring Shared Knowledge in Continual Learning. *ICML*, 2024.\
> [3]. Haowei Lin, et.al. Class incremental learning via likelihood ratio based task prediction. *ICLR*, 2024.\
> [4]. Md. Sazzad Hossain, et.al. Rethinking Task-Incremental Learning Baselines. *ICPR*, 2022.\
> [5]. Relja Arandjelovic, et.al. Look, listen and learn. *ICCV*, 2017.

---

### Official Review · Reviewer_Vqy2 · 2025-03-23

**Overall Recommendation:** 3

**Summary:**

In this paper, the author proposed probabilistic group mask selection, which aims to group parameters and explore the dependencies between them. In addition, the author used gumbel-softmax to make the sampling differentiable. To verify the method, the author conducted experiments on multiple datasets.

**Claims And Evidence:**

The author's claim is consistent with his technique and motivation. The main purpose is to explore the application of parameter isolation in the field of continual learning. More precisely, it is to explore the application of pruning in continual learning.

**Essential References Not Discussed:**

None.

**Experimental Designs Or Analyses:**

The experimental datasets selection is more in line with the measurement standards in the field. But the experimental effect does not seem to be very significant? As the sparsity index is significant, so such experimental results are acceptable. My question is that it seems not uncommon to generate a group of different masks in continual learning, so I am a little skeptical about the novelty of doing so.

**Methods And Evaluation Criteria:**

Yes, the proposed methods and techniques are all related to conditional masks, including mask selection, optimization and initialization.  However, I don’t see the specific application of dependency in the whole method. The more obvious point is that the final differentiable mask is generated by weighting different masks and task-informed mask initialization. But I’m not sure if this is the dependency discussed by the author, and I hope the author can explain this part.

**Other Comments Or Suggestions:**

As mentioned above, the issues of novelty and dependency need to be explained by the author.

**Other Strengths And Weaknesses:**

Ablation experiments are quite sufficient.

**Questions For Authors:**

None.

**Relation To Broader Scientific Literature:**

It is helpful for parameter isolation methods in continual learning.

**Theoretical Claims:**

Since it is an urgent review, I took a quick look at it. If there are any questions later, I will add them.

---

> ### Author Rebuttal · Authors · 2025-03-31
>
> Thank you for your valuable comments.
>
> **Clarifying the Role of Dependency in Our Method:** Modeling parameter dependency during subnetwork selection is the key contribution of this work, enabling more effective parameter allocation in incremental learning. Specifically, this involves two key aspects: (1) **Dependency** refers to the notion that the importance of a parameter should be evaluated in the context of its interactions with other parameters [1], in order to better reflect the collective contribution of parameter subsets. (2) To implement this idea, we partition parameters into groups and learn a categorical distribution over all possible selection combinations within each group, enabling joint evaluation of parameter subsets and thus capturing intra-group dependency. \
> In addition to dependency modeling, we introduce **task-informed mask initialization** to initialize task-specific mask distributions by leveraging similarities with prior task masks, thereby promoting efficient parameter reuse while preserving task-specific adaptability.
>
> **Clarifying the Novelty of Dependency-Aware Subnetwork Selection:** Parameter-isolation based incremental learning methods aim to assign compact yet effective subnetworks to individual tasks, thereby reducing capacity overhead and mitigating forgetting. However, existing approaches typically assume parameter independence, selecting parameters based on weight magnitude [2] or learnable importance scores [3], without considering how the importance of one parameter may depend on others. This assumption can lead to inaccurate parameter importance estimation, ultimately resulting in suboptimal subnetworks. To address this limitation, we propose a dependency-aware approach, Probabilistic Group Masking (PGM), which explicitly models parameter dependency during subnetwork selection. Specifically, (1) At the methodological level, PGM partitions parameters into multiple groups and evaluates all possible combinations within each group to capture intra-group dependency. Based on this modeling, PGM performs probabilistic sampling within each group to generate task-specific masks, with the entire process made differentiable via Gumbel-Softmax reparameterization. This design enables more dependency-aware subnetwork construction by jointly evaluating parameter combinations, thereby facilitating the selection of higher-quality subnetworks and enhancing the reuse of previously activated parameters. (2) At the theoretical level, we provide a theoretical analysis showing that modeling parameter dependency significantly reduces the risk of selecting suboptimal parameters. (3) At the empirical level, we conduct extensive experiments, including comparisons with state-of-the-art baselines, ablation studies of key components, and analysis of group size. The results consistently demonstrate that our dependency-aware approach reduces overall parameter usage while maintaining or even improving task performance.
>
> **References**:
> [1] Denis Kuznedelev, et al. CAP: Correlation-Aware Pruning for Highly-Accurate Sparse Vision Models. *NeurIPS*, 2023.
> [2] Arun Mallya, et al. PackNet: Adding Multiple Tasks to a Single Network by Iterative Pruning. *CVPR*, 2020.
> [3] Haeyong Kang, et al. Forget-free Continual Learning with Winning Subnetworks. *ICML*, 2022.

---

### Decision · Program_Chairs · 2025-05-01

**Decision:**

Accept (poster)

**Comment:**

The reviewers saw both strengths and weaknesses in the submitted version of the paper. After some discussion, the overall verdict was that this submission could be accepted in ICML-2025 while the concerns raised by reviewers should be addressed in the final version.